# Achievements and Perspectives in Metal–Organic Framework-Based Materials for Photocatalytic Nitrogen Reduction

Linkun Fan [1], Qin Yu [1], Jiazhen Chen [1], Usman Khan [1,2] , Xusheng Wang [1,2,*] and Junkuo Gao [1,2,*]

[1] Institute of Functional Porous Materials, School of Materials Science and Engineering, Zhejiang Sci-Tech University, Hangzhou 310018, China
[2] Zhejiang Provincial Innovation Center of Advanced Textile Technology, Shaoxing 312000, China
* Correspondence: xswang@zstu.edu.cn (X.W.); jkgao@zstu.edu.cn (J.G.)

**Abstract:** Metal–organic frameworks (MOFs) are coordination polymers with high porosity that are constructed from molecular engineering. Constructing MOFs as photocatalysts for the reduction of nitrogen to ammonia is a newly emerging but fast-growing field, owing to MOFs' large pore volumes, adjustable pore sizes, controllable structures, wide light harvesting ranges, and high densities of exposed catalytic sites. They are also growing in popularity because of the pristine MOFs that can easily be transformed into advanced composites and derivatives, with enhanced catalytic performance. In this review, we firstly summarized and compared the ammonia detection methods and the synthetic methods of MOF-based materials. Then we highlighted the recent achievements in state-of-the-art MOF-based materials for photocatalytic nitrogen fixation. Finally, the summary and perspectives of MOF-based materials for photocatalytic nitrogen fixation were presented. This review aims to provide up-to-date developments in MOF-based materials for nitrogen fixation that are beneficial to researchers who are interested or involved in this field.

**Keywords:** Haber–Bosch process; solar energy; photocatalytic nitrogen fixation; metal–organic frameworks

## 1. Introduction

Nitrogen is a necessary element to sustain life. It is abundant in the Earth's atmosphere in the form of virtually inert dinitrogen ($N_2$) gas that most organisms cannot metabolize [1]; instead, "fixed" forms such as ammonia ($NH_3$) can be metabolized. Before the widespread use of the Haber–Bosch process in the 1950s, fixed nitrogen was mainly derived from natural nitrogen fixation that involved geochemical processes such as lightning, and biological processes that used nitrogenase [2]. The Haber–Bosch process is a traditional and primary means for artificial nitrogen fixation, which converts $N_2$ and $H_2$ to $NH_3$ in the presence of a catalyst (Equation (1)). Now, nitrogen fixation via the Haber–Bosch process has exceeded natural nitrogen fixation. Nearly half of the existing human population will not exist without nitrogen fertilizers from the Haber–Bosch process. However, the Haber–Bosch process requires the expensive raw material hydrogen, intensive energy, and harsh conditions, including high temperatures (400–500 °C) and pressures (20–50 MPa). Therefore, finding a mild and green artificial nitrogen fixation method to replace or replenish the Haber–Bosch process is urgent.

$$N_2(g) + 3H_2(g) \rightarrow 2NH_3(g) \tag{1}$$

In recent years, photocatalysis technologies have been widely used in various fields [3–10]. With the development of photocatalysts, photocatalysis technology also makes the production of ammonia in mild conditions possible [11]. Currently, several

important homogeneous and heterogeneous photocatalysts for nitrogen fixation are available, such as Mo complex [12,13], Fe complex [14], $TiO_2$ [15–19], ZnO [20], $La_2TiO_5$ [21], $KNbO_3$ [22], Ti-zeolite [23], BiOBr [24,25], $Fe_2O_3$ [26], and g-$C_3N_4$ [27]. However, classical organometallic complexes (homogeneous catalysts) have not found practical application due to their low yields (around 10 equivalents of ammonia per equivalent of the catalyst), rapid catalyst deactivation, and the high costs of those catalysts. Thus, homogeneous catalysts for ammonia synthesis are currently the most useful for understanding nitrogenase. Although the stability has dramatically increased for such heterogeneous catalysts, their catalytic performance is still unsatisfactory, especially in the visible light region; most of those photocatalysts suffer from a narrow light absorption range, rapid photogenerated electron–hole combination, and a lack of rich catalytic sites.

Metal–organic frameworks (MOFs) are constructed by inorganic metal ions/metal clusters and multidentate organic ligands [28]. Since MOF-5, a highly porous crystalline material constituted by terephthalic acid (BDC) ligands and $Zn_4O$ clusters, was reported in Nature by Yaghi in 1999, MOFs began to develop rapidly [29]. The next two decades witnessed intensive efforts by numerous researchers to reveal new structures and applications of MOFs [30–39]. The naming of MOFs also has certain rules, mainly in the following four aspects: material composition (i.e., metal–organic framework, abbreviated as MOF) [29], structure (i.e., zeolitic imidazolate framework, abbreviated as ZIF) [40], function (i.e., multivariate metal–organic framework, abbreviated as MTV-MOF) [41], and the name of the laboratory or university (i.e., Materials of Institute Lavoisier, abbreviated as MIL) [33]. This naming method was subsequently adopted by most researchers [42–46]. Owing to their specific structural features such as large specific surface areas, high porosity, well-defined crystallinity, and increased numbers of active sites, MOFs have been widely used in gas adsorption and separation [47,48], fluorescence [49], sensing [44,45,50–52], ion conductivity [53], optoelectronics [54], thermal catalysis [55–59], electrocatalysis [60,61], photocatalysis [41,62–64], and so on. As photocatalysts, MOFs have many advantages: (i) the large specific surface area and highly ordered pore structure contribute to the mass transfer of reactants; (ii) the adjustable ligands make MOFs possess the ability to harvest light in a wide range; (iii) introducing defects into MOFs can expose more active sites, enhancing their nitrogen fixation activity [65–68]. Thus far, compared with photocatalytic hydrogen production and photocatalytic carbon dioxide ($CO_2$) reduction, the study of photocatalytic nitrogen fixation by MOFs is still in its initial stages of development.

This review firstly compared the different ammonia detection methods and the synthetic methods of MOF-based materials. The recent achievements in state-of-the-art MOF-based materials for photocatalytic nitrogen fixation have been further summarized. Finally, the summary and perspectives of MOF-based materials for photocatalytic nitrogen fixation were presented (Figure 1). This review aims to provide up-to-date developments in MOF-based materials for photocatalytic nitrogen fixation that are beneficial to researchers who are interested or involved in this field.

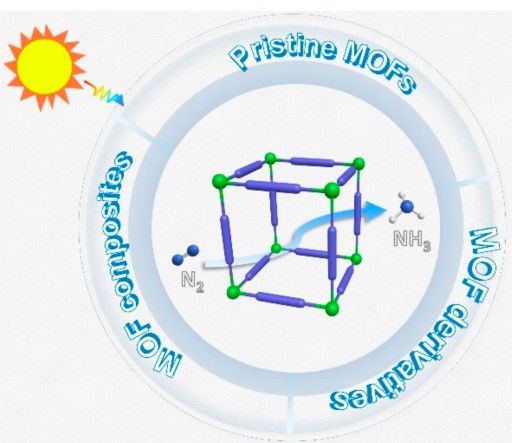

**Figure 1.** MOF-based materials for photocatalytic nitrogen fixation.

## 2. Ammonia Determination Methods

Ammonia is the main product of photocatalytic nitrogen fixation. It is typically detected with spectrophotometry (or colorimetry, including Nessler's reagent method, indophenol blue method, salicylic acid method, etc.), ion chromatography, $^1$H NMR (nuclear magnetic resonance) spectroscopy, fluorescence, $NH_4^+$ ion-selective electrodes, etc. All of the above-mentioned methods are accurate over a suitable range of ammonia concentrations for simple solutions of ammonia in water. However, the yields of ammonia are easily exaggerated or abated in the presence of interferants (impurity ions, impurity organic molecules, sacrificial agents, some solvents, etc.) or changes in pH, especially at low concentrations of $NH_3$ [69]. For example, sacrificial agents such as triethanolamine, methanol, ethanol, and other alcohols are widely used in photocatalysis for enhancing photogenerated electron–hole separation. Those $NH_x$ species that contain sacrificial agents, alcohols, and their oxidation products, also react with ammonia and disturb the quantification of results. Some metal ions may also leak from catalysts, resulting in inaccurate measurements of ammonia. Herein, we summarized the commonly used methods for ammonia detection and compared their advantages and limitations, guiding the selection of suitable detection methods for different photocatalytic systems.

Nessler's reagent method is based on the reaction of $Hg^{2+}$ and $I^-$ with ammonia under a strong alkaline condition to form a reddish-brown complex with a strong absorption peak near 420 nm (Equation (2)). The intensity of the absorption peak is proportional to the ammonia concentration in the absence of interferants. Nessler's reagent method can work well in natural conditions, as well as in acidic or alkali conditions (pH range from 1 to 13). However, metal ions such as $Ru^{3+}$, $Fe^{2+}$, $In^{3+}$, and $Ni^{2+}$ have obvious interference effects on ammonia detection, while $Ag^+$, $Ce^{3+}$, $Zn^{2+}$, $Cr^{3+}$, $Cu^{2+}$, and $Co^{2+}$ have negligible effect. Sacrificial agents and their oxidates, including methanol, formaldehyde, formic acid, ethanol, etc., have significant influences on the color after reaction, resulting in inaccurate estimations of ammonia (e.g., overestimated by 53 times in the presence of formaldehyde). It should be noted that mercury ions in Nessler's reagent are toxic, and should be used carefully. The shelf time of Nessler's reagent is also short (3 to 4 weeks) and requires frequent re-preparation. The reaction time of Nessler's reagent with ammonia also affects the accurate quantification of results, and a reaction time of 10 to 30 min is highly recommended [69].

$$NH_3 + 2[HgI_4]^{2-} + 3OH^- \rightarrow NH_2Hg_2OI\downarrow + 7I^- + 2H_2O \qquad (2)$$

The indophenol blue method involves the reaction of ammonia with phenol and hypochlorite in an alkaline solution to form indophenol blue, with strong absorption at around 700 nm (Scheme 1). Citrate buffer was often used to stabilize the pH of the reaction solution, and sodium nitroprusside was used as a catalyst to enhance the color

change during the reaction [70]. Compared with Nessler's reagent, the acidic condition will significantly reduce the accuracy of the indophenol blue method for ammonia detection. Furthermore, the indophenol blue method is not suitable for ammonia detection in the presence of sacrificial agents. The reaction time of phenol and hypochlorite with ammonia also affects the accurate quantification of results, and a reaction time of 60 min is highly recommended.

**Scheme 1.** The involved reaction of indophenol blue method.

The salicylic acid method is based on the reaction among sodium salicylate, sodium hypochlorite, and ammonia under alkali conditions using sodium nitroferricyanide as the catalyst (Scheme 2). The produced blue-colored compound with absorption at around 670 nm is the principle of this method [71]. Compared with the indophenol blue method, the salicylic acid method replaces phenol with sodium salicylate to avoid the formation of toxic volatile substances. Besides this, the salicylic acid method possesses all the advantages and limitations of the indophenol blue method.

**Scheme 2.** The involved reaction of salicylic acid method.

The ion chromatography method uses an ion exchange resin as the stationary phase to fill the chromatographic separation column, and the solution is used as the mobile phase for elution. Due to the different affinities between analytes and the ion exchange resin, the analytes pass through the chromatographic separation column at different speeds, resulting in separation and analysis. When the cation is analyzed, the cation exchange column is filled in the separation tower. On the contrary, an anion exchange column is used for anion detection [72]. Compared with the spectrophotometry methods, ion chromatography is a high-performance method with obvious advantages such as lower detection limits, reliable results, etc. The operation is also simple, and the ion chromatography method is highly recommended for ammonia detection. However, the high costs of the equipment mean that many research groups cannot use it for convenient analyses.

During photocatalytic nitrogen fixation, the source of ammonia in the solution is often controversial. Various pollution sources such as the catalyst itself, the water used as a solvent, the sacrificial agent, and contaminated $N_2$ or equipment may lead to false positive results. The $^{15}N_2$ isotope labeling experiment, using $^{15}N_2$ instead of $^{14}N_2$ as a gas source, is an important method to verify the ammonia source in solution, which can effectively avoid the interference of other N pollution. At the core of this method is that $^1H$ NMR spectroscopy can easily distinguish between $^{14}NH_4^+$ and $^{15}NH_4^+$, which is a powerful technique to confirm the source of detected ammonia. Therefore, with the help of isotope tracing technology, $^{15}N_2$ is often used as the nitrogen source to trace the source of detected ammonia.

In summary, the different methods have their advantages and drawbacks. Spectrophotometry methods, including Nessler's reagent method, indophenol blue method, and the salicylate method, are easily and affordably carried out, but are often disturbed by metallic ions, organics, and the pH of the solvent. The ion chromatography method is highly recommended for ammonia detection due to its lower detection limits, reliable results, etc. However, its high costs mean that many researchers cannot access it easily. The NMR method is often combined with isotope tracing technology to confirm the source of produced ammonia, but is rarely used for the quantitative analysis of ammonia.

## 3. Synthesis of MOF-Based Materials

The synthesis of MOF-based materials with high specific surface areas, porosities, and strong chemical/physical stabilities, plays a crucial role in photocatalytic nitrogen fixation. Such advanced physical structures of MOFs are clearly affected by synthesis methods, which furthermore determine their photocatalytic properties. The synthesis methods of pure MOFs include hydrothermal/solvothermal synthesis, microwave-assisted heating, slow evaporation, chemical vapor deposition (CVD), diffusion, electrochemical and mechanochemical synthesis methods, etc. A rational combination with other functional materials (graphene, g-$C_3N_4$, etc.) could form advanced MOF composites due to the synergistic effects of multifunctional units that enhance the performance of MOFs. Moreover, MOFs were also explored as precursors for preparing a range of MOF-derived materials from carbon- and metal-based materials [73]. Here, we systematically introduce the synthesis method of MOFs.

The hydrothermal/solvothermal method involves preparing materials by dissolving and recrystallizing powder in a sealed pressure vessel, using water/organic solvent as the solvent. The reaction in the sealed pressure vessel is under subcritical or supercritical conditions. This method can obtain powder MOF crystallite products, and even crystal products that are suitable for single-crystal analysis. The reason for this is mainly that moderate temperature and pressure can promote the complete dissolution of reactants in solvents. MOFs that are synthesized by the hydrothermal/solvothermal method usually have high thermal stability [29]. However, this method cannot overcome the mixture of different compound crystals in the product, which makes separation very difficult. Moreover, the reaction time is long and the energy consumption is high. It should be noted that the hydrothermal/solvothermal method is the most commonly used method for MOF synthesis.

Compared with the hydrothermal/solvothermal method, the microwave-assisted heating method is more energy- and time-saving [74]. Microwave-assisted heating has an internal thermal effect, and the temperature of the entire reaction system is uniform. Under the function of a high-frequency magnetic field, the microwave-assisted heating method can produce a rapid heat effect.

The slow evaporation method refers to a process that is carried out as follows: firstly, a mixture of metal salts, organic ligands, and water/organic solvents is fully stirred to obtain a clear solution; then, the solution is stood for several days to evaporate the solvent slowly; finally, the required product is obtained [75]. This route is sometimes preferred because it is a room-temperature method; it is suitable for growing large single crystals, but requires a long time.

Chemical vapor deposition (CVD) is based on the reaction of vaporized substances on or nearby a support to form a uniform thin film. The thickness of the formed film can be well controlled. Compared with the CVD method, the conventional solvent-based procedures are incompatible with nanofabrication because of corrosion and contamination risks. The CVD method is suitable for microelectronics production. For example, Stassen et al. reported a CVD approach to produce high-quality films of ZIF-8 [76]. The thickness of ZIF-8 film prepared by the CVD method is uniform and controllable, significantly promoting the application of MOFs in microelectronics.

The diffusion method refers to the separation of organic ligands from metal ions in different solvents, and then reacting them with each other through slow diffusion. The diffusion method involves liquid diffusion [77], gel diffusion, and gas diffusion [78]. This method is applied as a result of the poor solubility of the coordination products. Directly mixing organic ligands and metal ions will generally form powder products, and the solubility of these products is poor. The reaction rates and the yields are low, and large quantities of products cannot be obtained using this method.

The principle behind the electrochemical synthesis of MOFs is that metal ions are continuously provided by anodic oxidation, replacing metal salts as metal sources. Then, metal ions are reacted with dissolved ligand molecules in conductive media to realize

the coordination and generation of MOF crystals. Electrochemical synthesis methods mainly include the anode synthesis method, cathode synthesis method, indirect bipolar electrodeposition method, electroplating replacement method, and the electrophoretic deposition method. For example, Ameloot et al. demonstrated an electrochemical method as being a feasible and efficient way of growing [Cu$_3$(BTC)$_2$] coatings [79]. This method makes it possible for MOFs to use standard lithography combined with electrochemical synthesis as a functional coating method in the manufacture of microelectronic devices. MOFs synthesized by electrochemical methods possess high porosity, and the synthesis speed is rapid. Electrochemical synthesis methods can continuously synthesize controllable particle morphologies under mild reaction conditions; however, this method's yield is low and easily produces by-products.

The mechanochemical synthesis method refers to the direct reaction between metal ions and organic ligands through mechanical grinding. This method dramatically reduces pollution and costs, which makes mass production of MOFs possible. For example, Friščić et al. constructed a MOF named [Zn$_2$(ta)$_2$(DABCO)] using an improved mechanochemical approach, designated ion- and liquid-assisted grinding (ILAG) [80]. This is the first example of using anion templating in mechanosynthesis, and of using additives to enhance MOF mechanosynthesis. Recently, Samal et al. presented a simple mechanochemical synthesis of MOFs using a kitchen grinder [81]. This tool successfully synthesized multi-gram-scale MOFs including ZIF-8, CuBTC, and MIL-100(Fe).

## 4. MOF-Based Materials for Photocatalytic Nitrogen Fixation

Due to their large pore volumes, adjustable pore sizes, wide light harvesting range, and high densities of exposed catalytic sites, MOFs have been explored for photocatalytic nitrogen fixation. However, it is still in its preliminary stages of development. In addition to pure MOFs, advanced MOF composites and MOF derivatives have also been studied for photocatalytic nitrogen fixation (Table 1).

**Table 1.** Photocatalytic nitrogen fixation by MOF-based materials.

| Photocatalyst | Light Source | Sacrificial Agents | NH$_3$ Yield | AQY/% | Ref. |
|---|---|---|---|---|---|
| | | Pure MOFs | | | |
| NH$_2$-MIL-125 (Ti) | Xe Lamp (300 W, L40) | None | 12.25 $\mu$molg$^{-1}$ h$^{-1}$ | 0.26 (400 nm) | [82] |
| OH-MIL-125 (Ti) | Xe Lamp (300 W, L40) | None | 5.04 $\mu$molg$^{-1}$ h$^{-1}$ | / | [82] |
| CH$_3$-MIL-125 (Ti) | Xe Lamp (300 W, L40) | None | 1.39 $\mu$molg$^{-1}$ h$^{-1}$ | / | [82] |
| UiO-66-UV-Vis | Xe Lamp (300 W, UV-vis) | None | 256.60 $\mu$molg$^{-1}$ h$^{-1}$ | / | [83] |
| UiO-66(SH)$_2$-200 | Xe Lamp (300 W, L40) | None | 32.40 $\mu$molg$^{-1}$ h$^{-1}$ | 0.45 (420 nm) | [84] |
| MIL-101(Fe) | Xe Lamp (300 W, full-spectrum) | None | 50.36 $\mu$molg$^{-1}$ h$^{-1}$ | / | [85] |
| MIL-100(Fe) | Xe Lamp (300 W, full-spectrum) | None | 46.53 $\mu$molg$^{-1}$ h$^{-1}$ | / | [85] |
| MIL-88(Fe) | Xe Lamp (300 W, full-spectrum) | None | 40.04 $\mu$molg$^{-1}$ h$^{-1}$ | / | [85] |
| MIL-53(Fe$^{II}$/Fe$^{III}$)-0.1 | Xe Lamp (300 W, L42) | K$_2$SO$_3$ | 306.00 $\mu$molg$^{-1}$ h$^{-1}$ | 0.12 (420 nm) | [86] |
| Al–PMOF(Fe) | Xe Lamp (100 mWcm$^{-2}$, L42) | CH$_3$OH | 7.06 $\mu$molg$^{-1}$ h$^{-1}$ | / | [87] |

**Table 1.** *Cont.*

| Photocatalyst | Light Source | Sacrificial Agents | NH$_3$ Yield | AQY/% | Ref. |
|---|---|---|---|---|---|
| MOF-76(Ce) | Xe Lamp (300 W, full-spectrum) | None | 34.20 $\mu$molg$^{-1}$ h$^{-1}$ | / | [88] |
| Gd-IHEP-8 | Xe Lamp (300 W, AM 1.5 G filter) | None | 220.00 $\mu$molg$^{-1}$ h$^{-1}$ | 2.25 (365 nm) | [89] |
| Gd-IHEP-7 | Xe Lamp (300 W, AM 1.5 G filter) | None | 128.00 $\mu$molg$^{-1}$ h$^{-1}$ | 1.72 (365 nm) | [89] |
| U(0.5Hf) | Xe Lamp (300 W, full-spectrum) | K$_2$SO$_3$ | 351.80 $\mu$molg$^{-1}$ h$^{-1}$ | 0.1 (420 nm) | [90] |
| U(0.5Hf)-2SH | Xe Lamp (300 W, L42) | K$_2$SO$_3$ | 116.10 $\mu$molg$^{-1}$ h$^{-1}$ | 0.55 (420 nm) | [90] |
| NU6(Ce–Hf) | Xe Lamp (300 W, full-spectrum) | K$_2$SO$_3$ | 158.4 $\mu$molg$^{-1}$ h$^{-1}$ | 0.65 (380 nm) | [91] |
| MOF composites | | | | | |
| ZIF-67@PMO$_{12}$ | Xe Lamp (300 W, full-spectrum) | C$_2$H$_5$OH | 39.40 $\mu$molg$^{-1}$ h$^{-1}$ | / | [92] |
| ZIF-67@PMO$_{11}$V | Xe Lamp (300 W, full-spectrum) | C$_2$H$_5$OH | 70.00 $\mu$molg$^{-1}$ h$^{-1}$ | / | [92] |
| ZIF-67@PMO$_{10}$V$_2$ | Xe Lamp (300 W, full-spectrum) | C$_2$H$_5$OH | 74.80 $\mu$molg$^{-1}$ h$^{-1}$ | / | [92] |
| ZIF-67@PMO$_9$V$_3$ | Xe Lamp (300 W, full-spectrum) | C$_2$H$_5$OH | 134.60 $\mu$molg$^{-1}$ h$^{-1}$ | / | [92] |
| ZIF-67@PMO$_4$V$_8$ | Xe Lamp (300 W, full-spectrum) | C$_2$H$_5$OH | 149.00 $\mu$molg$^{-1}$ h$^{-1}$ | / | [92] |
| Au@UiO-66 | Xe Lamp (300 W, L42) | None | 18.90 $\mu$molg$^{-1}$ h$^{-1}$ | 1.54 (520 nm) | [93] |
| MOF-74@C$_3$N$_4$ | Xe Lamp (300 W, L40) | CH$_3$OH | 330.00 $\mu$molg$^{-1}$ h$^{-1}$ | / | [94] |
| MIL-125@TiO$_2$ | Xe Lamp (300 W, 200 mWcm$^{-2}$) | None | 102.70 $\mu$molg$^{-1}$ h$^{-1}$ | / | [95] |
| GSCe (Graphene@Ce-UiO-66) | LED (6 W, 365 nm) | None | 110.24 $\mu$molg$^{-1}$ h$^{-1}$ | 9.25 (365 nm) | [96] |
| 9MX-MOF | Xe Lamp (300 W, full-spectrum) | Na$_2$SO$_3$ | 88.79 $\mu$molg$^{-1}$ h$^{-1}$ | / | [97] |
| MOF derivatives | | | | | |
| Ru–In$_2$O$_3$ HPNs | Xe Lamp (300 W, AM 1.5 G filter) | CH$_3$OH | 44.50 $\mu$molg$^{-1}$ h$^{-1}$ | / | [98] |

*4.1. Pristine MOFs*

Although MOFs have been studied for more than 30 years, they have only been used for photocatalytic nitrogen fixation since 2019. Herein, we systematically summarize the state-of-the-art MOFs for photocatalytic nitrogen fixation based on metal nodes: transition metal- and post-transition metal-based MOFs.

4.1.1. Transition Metal-Based MOFs

The unoccupied and occupied d-orbitals in some transition metals-based MOFs have appropriate energy and symmetry, which makes these MOFs effectively adsorb and activate N$_2$. The empty *d* orbital of open metal sites in MOFs can accept the electrons from the occupied $\sigma$ orbital of N$_2$. At the same time, the occupied *d* orbitals of open metal sites donate electrons to the empty $\pi$* orbital of N$_2$. The back donated bonds not only weaken

N≡N, but also strengthen the metal-nitrogen bond [99], making transition metals-based MOFs for photocatalytic nitrogen fixation possible.

Cerium-Based MOFs

Cerium, with its electron configuration of $[Xe]4f^26s^2$, exhibits flexible valence transformation behavior between $Ce^{3+}$ and $Ce^{4+}$, with occupied $4f^1$ and unoccupied $4f^0$ orbitals, respectively. On this basis, the cerium with empty orbits and filled orbitals can mimic $\pi$ back donation for further catalysis [100].

Recently, Zhang et al. applied MOF-76(Ce) to effectively transform $N_2$ into $NH_3$, mimicking $\pi$ back donation [88]. In this study, MOF-76(Ce) nanorods with Ce coordinate unsaturated sites (Ce-CUS) were prepared using the solvothermal method (Figure 2a). Under full-spectrum light source irradiation, the photogenerated electrons were first transferred to Ce-CUS sites, then to the $\pi$ antibonding orbital of adsorbed $N_2$ molecules on it. The electrons entering the $\pi$ antibonding orbital would weaken the N≡N bond, significantly improving photocatalytic nitrogen fixation efficiency. The photocatalytic nitrogen fixation stability of MOF-76(Ce) is comparable to $CeO_2$ (Figure 2b). Moreover, due to the synergy between Ce-CUS and $N_2$, the photocatalytic nitrogen fixation activity of MOF-76(Ce) is higher than $CeO_2$, with an activity of 34.2 $\mu molg^{-1} h^{-1}$. It should be noted that this is the first kind of study that involved using MOFs for nitrogen fixation.

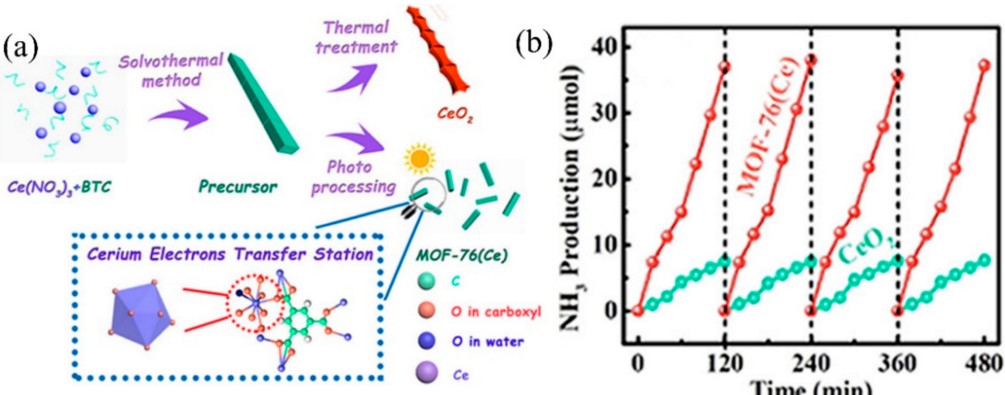

**Figure 2.** (**a**) Synthesis diagram of MOFs-76(Ce). (**b**) MOFs-76(Ce) stability test of photocatalytic nitrogen fixation. Reproduced with permission [88]. Copyright 2019, American Chemical Society.

Titanium Metal-Based MOFs

Ti-based MOFs have been widely explored in photocatalytic hydrogen production [101–103], photocatalytic $CO_2$ reduction [104,105], and environmental protection; however, photocatalytic nitrogen fixation is still in its preliminary stages of development.

Recently, our group initially synthesized three functional group-decorated isostructural MOFs ($NH_2$-MIL-125 (Ti), OH-MIL-125 (Ti), and $CH_3$-MIL-125 (Ti)) for visible-light-driven photocatalytic nitrogen fixation (Figure 3a) [82]. The introduced functional groups effectively enlarged the light-harvesting range of MIL-125 (Ti) from the ultraviolet (UV) region to the visible light region. After being exposed to visible light (400–800 nm) for about 15 h, the produced ammonia of $NH_2$-MIL-125 (Ti), $CH_3$-MIL-125 (Ti), and OH-MIL-125 (Ti) reached 183.76 $\mu molg^{-1}$, 20.88 $\mu molg^{-1}$, and 75.48 $\mu molg^{-1}$, respectively (Figure 3b). It should be noted that no sacrificial agent participated in the photocatalytic nitrogen fixation process for those MOFs. The reason for the highest nitrogen fixation rate of $NH_2$-MIL-125 (Ti) can be explained by the introduction of $NH_2$, which significantly increased light absorption to 550 nm, and by the exposed Ti coordinational unsaturated sites induced by a linker defect (Figure 3c). The photocatalysis mechanism can be illustrated as follows: under visible light irradiation, photogenerated electrons in organic ligands are transferred to exposed Ti coordinational unsaturated sites and reduce $Ti^{4+}$ to $Ti^{3+}$; then, the electron

in Ti$^{3+}$ is further transferred to the π anti-bond of N$_2$ to weaken the strong N≡N bond, eventually reducing the N$_2$ to NH$_3$ (Figure 3d).

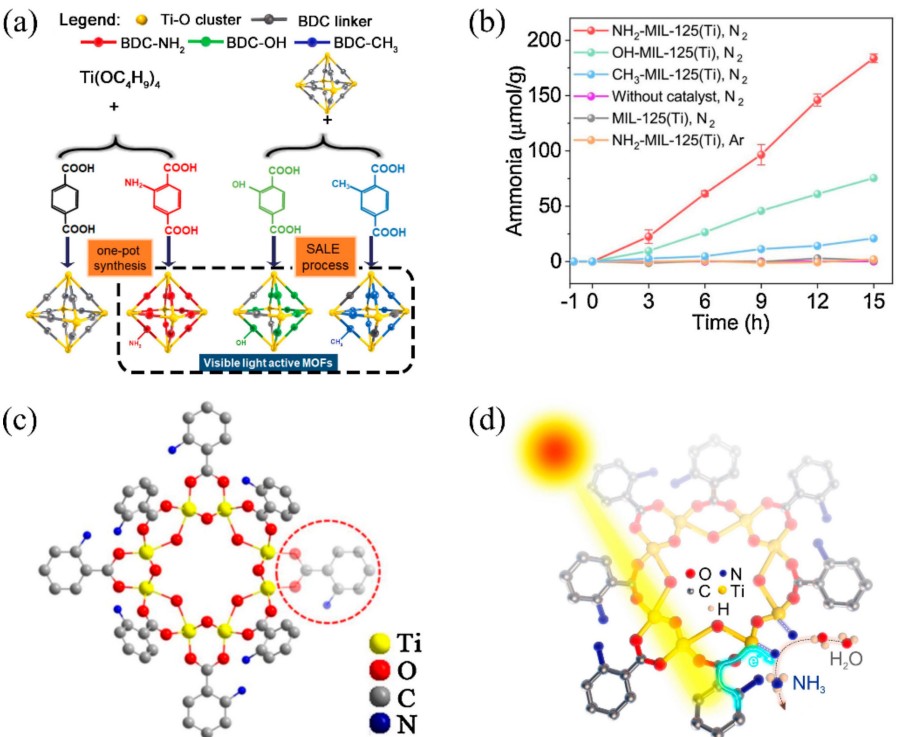

**Figure 3.** (**a**) Schematic diagram of isostructural MOFs with different functional groups for photocatalytic nitrogen fixation. (**b**) Curve of NH$_4^+$ content with time generated. (**c**) Schematic diagram of connection defects of NH$_2$-MIL-125(Ti). (**d**) Mechanism of NH$_2$-MIL-125(Ti) photocatalytic nitrogen fixation. Reproduced with permission [82]. Copyright 2020, Elsevier.

Zirconium-Based MOFs

Defective UiO-66(Zr) series MOFs reflect excellent application prospects in photocatalytic hydrogen production, pollutant degradation, and absorption, which result from their large surface areas and suitable pore structures [106–108].

Recently, UiO-66-UV-vis with linker defects induced by light was published by Gao et al. [83] to improve photocatalytic nitrogen fixation activity. In order to improve the increase in photocatalytic nitrogen fixation activity that was attributed to linker defect but not cluster defect, three different types of UiO-66, UiO-66-fresh, UiO-66-UV-vis, and UiO-66-PSE, were studied for photocatalytic nitrogen fixation. UiO-66-fresh, UiO-66-UV-vis, and UiO-66-PSE stand for fresh prepared UiO-66, UV-vis light treated UiO-66, and defect repaired UiO-66 through a post-synthetic ligand exchange process (PSE), respectively. Compared with UiO-66-fresh, the coordinated formic acid and acetic acid were removed in UiO-66-UV-vis after exposure to ultraviolet light. The exposed coordination of unsaturated metal sites greatly enhanced the activity of UiO-66-UV-vis up to 256.6 μmolg$^{-1}$ h$^{-1}$. The coordination unsaturated Zr node on UiO-66 can inject photogenerated electrons into the antibonding π-orbitals of N$_2$ to promote the activation and dissociation of N$_2$. In contrast, the exposed coordination unsaturated metal sites were recovered by the terephthalic acid linker in UiO-66-PSE, which resulted in lower photocatalytic activity, even lower than that of UiO-66-fresh.

Guo et al. also introduced defect Zr-based MOFs created by thermal treatment, using UiO-66(SH)$_2$-200 as the photocatalyst for the reduction of N$_2$ (Figure 4a) [84]. The Zr clusters were dehydrated by thermal treatment, thus providing accessible [Zr$_6$O$_6$] sites for N$_2$ adsorption and activation. The optical temperature was 200 °C. SH groups were also introduced into UiO-66 to improve the absorption edge of the light to visible light. UiO-

66(SH)$_2$-200 shows a photocatalytic nitrogen fixation activity up to 32.4 µmolg$^{-1}$ h$^{-1}$ under visible light. In-situ DRIFTS (diffuse reflectance infrared Fourier transform spectroscopy) revealed that the N$_2$ molecule was gradually reduced to an N$_x$H$_y$ intermediate and to NH$_3$, finally (Figure 4b) [109]. DFT (density functional theory) calculations further revealed that the photoelectron initiates the reduction of the N$_2$, immediately followed by the protonation of the activated N$_2$ (Figure 4c).

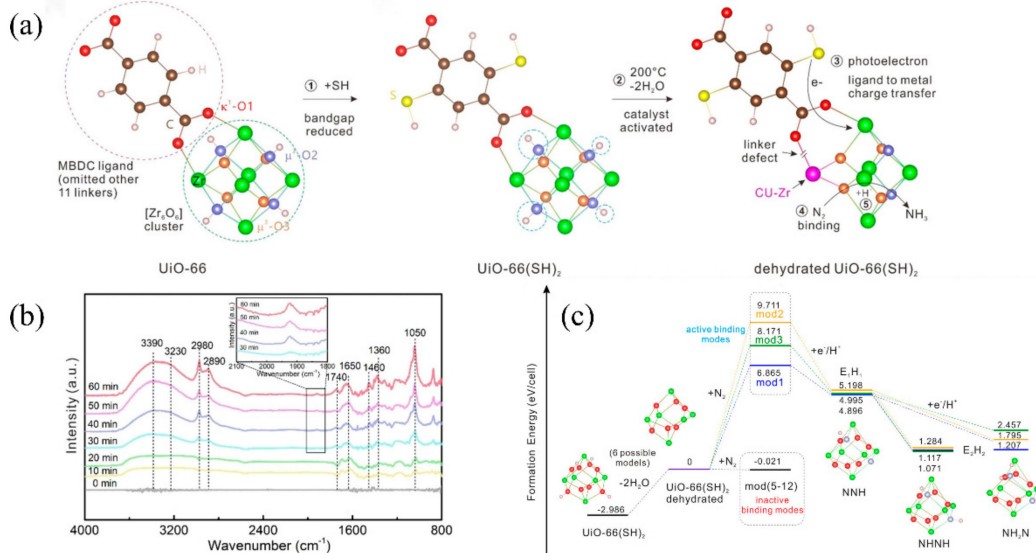

**Figure 4.** (**a**) Schematic evolution of UiO-66 to dehydrated UiO-66(SH)$_2$. (**b**) Time-dependent in situ DRIFT spectra of UiO-66(SH)$_2$-200. (**c**) The formation energy diagram for the different models simulating the N$_2$ reduction process. Reproduced with permission [84]. Copyright 2022, Wiley Online Library.

Iron-Based MOFs

Fe is an essential transition metal in nitrogenase, which plays a vital role in photocatalytic nitrogen fixation. There have been many types of research on reducing N$_2$ by utilizing the Fe active site [110–113]. When the Fe metal site is exposed for adsorbing N$_2$, the unpaired electrons of the *d* orbital in Fe will transfer to the π antibonding orbital of the N$_2$ molecule to form a strong Fe–N bond that weakens the N≡N bond.

Recently, in order to prove the importance of the transition metal Fe for photocatalytic nitrogen fixation [85], Li et al. studied the nitrogen fixation performance of MIL-101(Fe), MIL-100(Fe), MIL-88(Fe), and MIL-101(Cr). Notably, MIL-101(Fe) exhibited the highest photocatalytic nitrogen activity (50.36 µmolg$^{-1}$ h$^{-1}$), whereas isostructural MIL-101(Cr) had almost no activity. DFT calculations revealed that MIL-101(Fe) showed more electronic supply capacity, higher adsorption energy of N$_2$, and a lower reaction barrier than MIL-101(Cr), confirming the important role Fe has during photocatalytic nitrogen fixation (Figure 5).

Owing to its unique multi-iron metallocluster (Fe$^{2+}_3$Fe$^{3+}_4$M$^{3+}$, M = Mo, V, Fe), nitrogenase exhibits excellent nitrogen fixation activity (Figure 6a). Recently, Zhao et al. synthesized a MIL-53(Fe$^{2+}$/Fe$^{3+}$) containing both Fe$^{2+}$ and Fe$^{3+}$ for photocatalytic nitrogen fixation [86]. In this MOF, Fe$^{2+}$ and Fe$^{3+}$ simulated Fe$^{2+}$ activity sites and high valence metal ions (M = Mo, V, Fe) in nitrogenase (Figure 6b). The Fe$^{3+}$ in MIL-53(Fe$^{2+}$/Fe$^{3+}$) can be partly reduced into Fe$^{2+}$ by ethylene glycol (EG), and the Fe$^{2+}$/Fe$^{3+}$ ratio can be regulated from 0.18:1 to 1.21:1 by changing the EG content. Notably, when the ratio of Fe$^{2+}$/Fe$^{3+}$ was 1.06:1 (Figure 6c), photocatalytic nitrogen fixation activity reached its highest value of 306 µmolh$^{-1}$ g$^{-1}$.

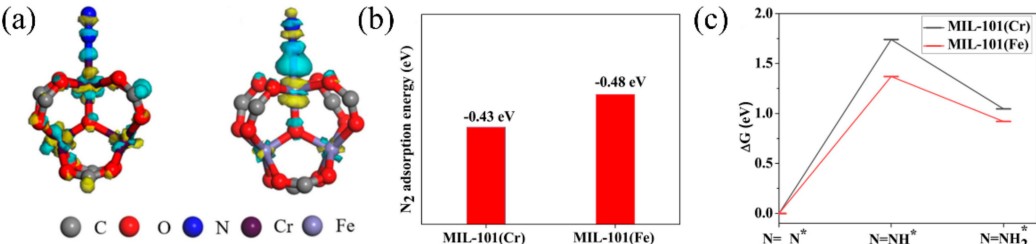

**Figure 5.** (**a**) Charge density difference of an $N_2$ molecule absorbed at MIL-100(Cr) and MIL-101(Fe). (**b**) Adsorption energies of $N_2$ on MIL-101(Cr) and MIL-101(Fe). (**c**) Calculated free energy diagram for reduction of $N_2$ of MIL-101(Cr) and MIL-101(Fe). Reproduced with permission [85]. Copyright 2020, Elsevier.

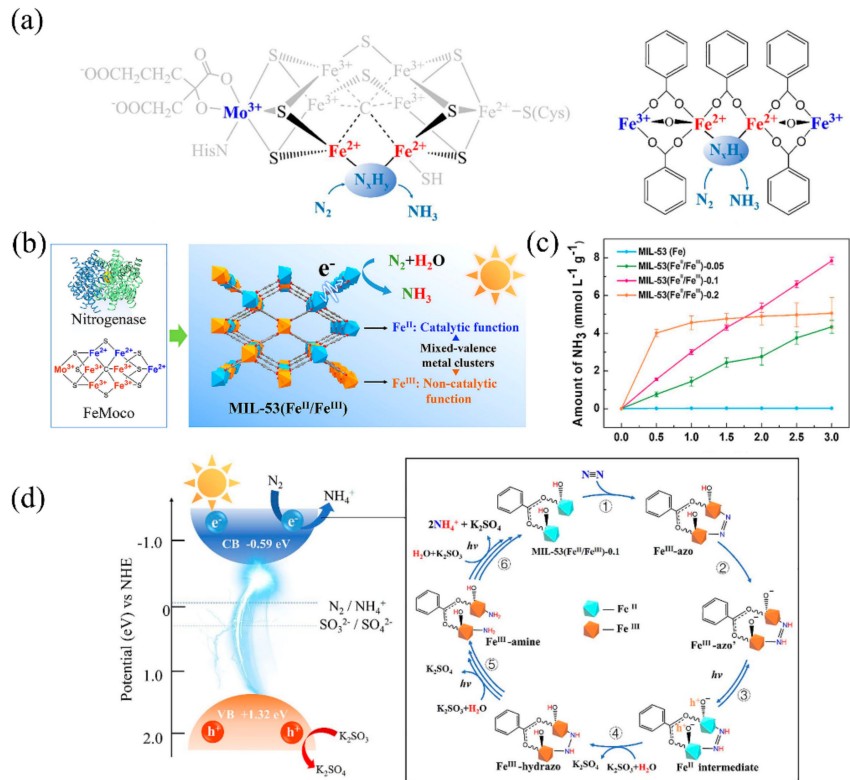

**Figure 6.** (**a**) The multi-iron metallocluster of nitrogenases and MIL-53($Fe^{2+}/Fe^{3+}$). (**b**) Comparison of nitrogenase MIL-53($Fe^{2+}/Fe^{3+}$) to realize the photocatalytic nitrogen fixation. (**c**) $NH_3$ yield of MIL-53($Fe^{2+}/Fe^{3+}$) with different $Fe^{2+}/Fe^{3+}$ ratios. (**d**) The proposed mechanism of photocatalytic fixation of $N_2$ on MIL-53($Fe^{2+}/Fe^{3+}$)-0.1 catalyst. Reproduced with the permission of the author [86]. Copyright 2020, Elsevier.

The proposed mechanism for photocatalytic nitrogen fixation by MIL-53($Fe^{2+}/Fe^{3+}$) can be divided into six steps. (i) $N_2$ adsorbs onto $Fe^{2+}$ active sites of MOF, in which the electrons transfer to $N_2$ to form the $Fe^{3+}$-azo intermediate. (ii) Then, the hydrogen transfer quickly occurs in the $Fe^{3+}$-azo intermediate. (iii) In the photo-excitation stage, holes transfer to the $Fe–O^-$ radical, and $Fe^{3+}$ reduces to $Fe^{2+}$ by electrons. (iv) The sacrificial agent $K_2SO_3$ eliminates the holes, and $Fe^{2+}$-hydrazo transforms into $Fe^{3+}$-hydrazo, with $H_2N–NH_2$ species appearing. (v) $Fe^{3+}$-hydrazo transfers to $Fe^{3+}$-amine with further hydrogenation. (vi) Finally, the $NH_3$ releases gradually, benefitting from continuous hydrogenation (Figure 6d).

Transitional Bimetallic MOFs

Similarly to MIL-53($Fe^{2+}/Fe^{3+}$), An et al. developed bimetallic MOFs containing Zr and Hf for photocatalytic nitrogen fixation. Zr simulates the $Fe^{2+}$ active site in nitrogenase, acting as the active site through the $\pi$ antibonding mechanism. Hf imitates the high valence metal in nitrogenase to promote electron transfer and utilization. An SH group was also introduced to extend the absorption edge to the visible light region [90]. As the SH group expands the absorption spectrum to 502 nm and the synergistic effect of Zr and Hf, U(0.5Hf)-2SH with 50% Hf exhibits extremely superior photocatalytic nitrogen fixation activity (116.1 $\mu molh^{-1}\ g^{-1}$) under visible light (Figure 7a). The constructed ligand-to-metal-to-metal electron transfer (LMMET) pathway in MOFs promotes the transferring and utilization of photogenerated electrons. Under visible light irradiation, the electrons move from the highest occupied molecular orbit (HOMO) to the lowest occupied molecular orbit (LOMO). At the same time, the holes are consumed by the sacrificial agent $K_2SO_3$, the electrons are transferred to the multi-metal clusters, and finally the $N_2$ is reduced to $NH_3$ (Figure 7b). Except for zirconium- and hafnium-based MOFs, An et al. further synthesized a stable amino-functionalized UiO-66 with bimetallic Ce-Hf nodes for photocatalytic nitrogen fixation [91]. In this MOF, the introduced $NH_2$ group expands the absorption edge, the Ce species acts as an electron buffer tank to enhance electron transfer, and the Hf species plays the part of active catalytic sites to improve the selectivity of the nitrogen fixation reaction. When the molar ratio of Ce-Hf is 1:1, the nitrogen fixation activity was the highest (158.4 $\mu molh^{-1}\ g^{-1}$) under visible light, with $K_2SO_3$ as the sacrificial agent.

4.1.2. Post-Transition Metal-Based MOFs

Compared with transition metal-based MOFs, the post-transition metal-based variations are rarely reported to reduce nitrogen to ammonia as photocatalysts.

Gadolinium-Based MOFs

Hu et al. developed two viologen-based radical-containing metal–organic frameworks, Gd-IHEP-7 and Gd-IHEP-8 [89]. A single-crystal-to-single-crystal (SCSC) transformation occurred from two-dimensional (2D) Gd-IHEP-7 to three-dimensional (3D) Gd-IHEP-8 when heating the Gd-IHEP-7 in the air at 120 °C. With a rearrangement of the $Gd^{3+}$ coordination environment, enhanced photocatalytic nitrogen fixation activity emerged with the SCSC transformation. Chemisorption of $N_2$ onto the catalytic sites is a pre-condition for photocatalytic nitrogen fixation. Both the adsorbed $N_2$ on active Gd metal sites for Gd-IHEP-7 and Gd-IHEP-8 were activated, evidenced by the elongated N–N bond length and the shortened Ga–N bond length. Compared with the distal (D) route, the alternative (A) route is more favorable for both Gd-IHEP-7 and Gd-IHEP-8. While even theoretical calculations indicate that similar photocatalytic nitrogen fixation pathways exist for both RMOF (Gd-IHEP-7, Gd-IHEP-8), the intermediates for Gd-IHEP-8 showed better stability, resulting in a better nitrogen fixation activity of 220 $\mu molh^{-1}\ g^{-1}$.

Aluminium-Based MOFs

The post-transition metals can not only function as active sites as in Gd-IHEP-8, but they also act as metal nodes that impart high framework stability to MOFs. Recently, Shang et al. developed two porphyrin-based metal–organic frameworks for photocatalytic nitrogen fixation, named Al-PMOF and Al-PMOF(Fe). Compared with Al-PMOF, Al-PMOF(Fe) not only has Al as the metal node to stabilize the framework of the MOF, but also has Fe as the active center to adsorb and reduce the $N_2$ to $NH_3$ [87]. The structure of Al-PMOF(Fe) is shown in Figure 8a. The atomically isolated Fe in Al-PMOF(Fe) was proven through X-ray absorption spectroscopy (XAS) and X-ray photoelectron spectroscopy (XPS). Photocatalytic nitrogen fixation experiments showed that Al-PMOF(Fe) had better activity than Al-PMOF under visible light, and that the produced $NH_3$ originated from $N_2$. It was confirmed that the addition of Fe as the active center effectively increased the adsorption of $N_2$ and further enhanced the photocatalytic performance (Figure 8b,c). DFT

calculations were further applied to establish the photocatalytic nitrogen fixation reaction pathway (Figure 8d–f). The first hydrogenation from $N_2^*$ to $N_2H^*$ showed no obvious difference between alternating and distal pathways, but further hydrogenation showed that the alternating pathway seemed more likely to occur. Notably, the release of $1NH_3$ in the distal process requires significant energy, making the reaction difficult.

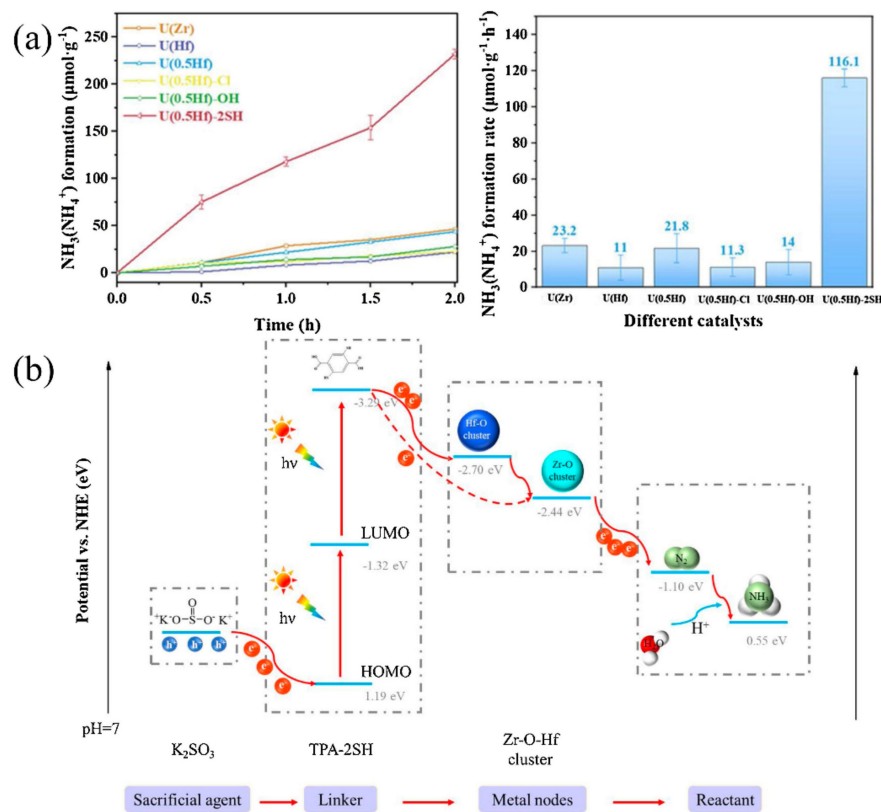

**Figure 7.** (**a**) Photocatalytic $NH_3$ yield of different MOFs. (**b**) Photocatalytic nitrogen fixation mechanism of U(0.5Hf)-2SH. Reproduced with permission [90]. Copyright 2021, Elsevier.

*4.2. MOF Composites*

Pristine MOFs can not only participate in photocatalytic reduction of $N_2$ to $NH_3$ directly, but they can be also combined with other functional materials (POMs, g-$C_3N_4$, metal nanoparticles, semiconductors, etc.) to enhance catalytic performance. The enhanced performance of MOF composites can be attributed to the synergistic effects of multifunctional units, which are widely used for catalysis. The functional materials can be inserted into the pores of MOFs, covered by the MOFs shell, or intimately connected with MOFs. In the MOF composites, MOFs can act as active sites, light harvesters, or porous scaffolds to support functional units, and enhance the mass transfer of $N_2$ and produce $NH_3$.

For example, Li et al. incorporated Keggin-type $PMo_{12-X}V_X$ (X = 0, 1, 2, 3, 8), a type of polyoxometalate (POM), into ZIF-67 to enhance the activity of photocatalytic nitrogen fixation [92]. In the advanced MOF composites, $PMo_{12-X}V_X$ acted as a light harvester, and the porous ZIF-67 adsorbed the $N_2$ and utilized the photogenerated electrons from $Pmo_{12-X}V_X$ for the photocatalytic reduction of $N_2$ to $NH_3$. The synergistic effect between POM and ZIF-67 greatly promoted the separation of photogenerated electrons and holes, resulting in enhanced photocatalytic performance of ZIF-67@$Pmo_4V_8$ to reach 149.0 $\mu molL^{-1}\,h^{-1}$. The proposed mechanism of photocatalytic nitrogen fixation can be divided into three steps. (i) ZIF-67 provide active sites to adsorb large amounts of $N_2$. (ii) Under the irradiation of visible light, the photogenerated electrons and holes separate in POMs; at the same time, the holes are consumed by sacrificial ethanol, resulting in reduced

POMs. (iii) The photogenerated electrons then transfer from POMs to ZIF-67, and finally move to the $N_2$ molecule to participate in the reduction of $N_2$.

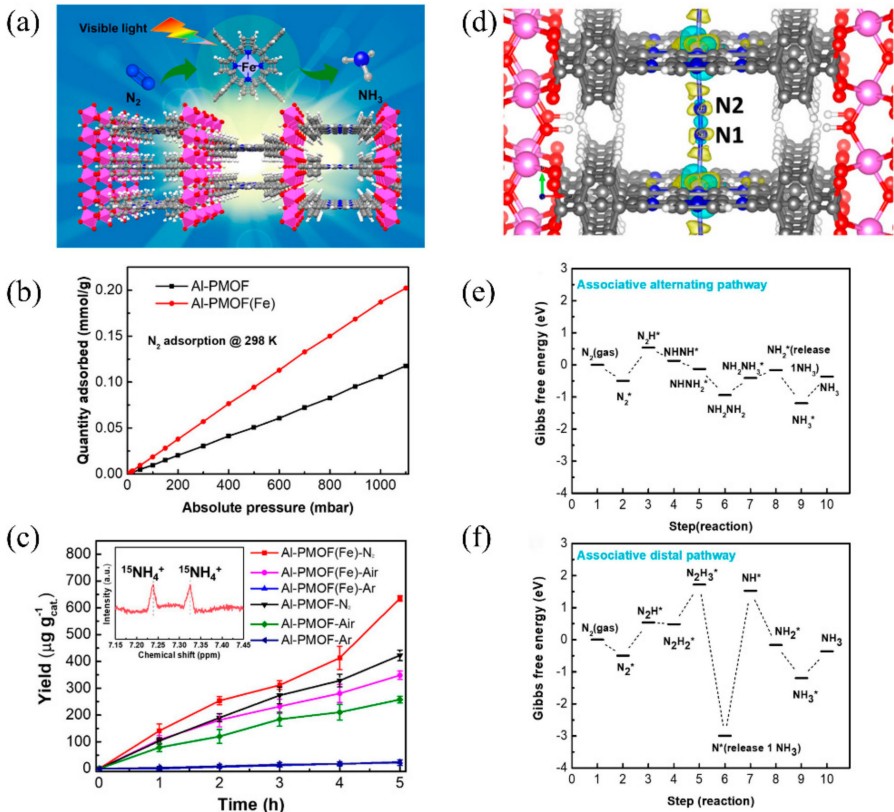

**Figure 8.** (**a**) The structure of Al-PMOF(Fe). (**b**) $N_2$ adsorption isotherms. (**c**) Photocatalytic nitrogen fixation activity of Al-PMOF(Fe) and Al-PMOF. (**d**) Charge different maps for Al-PMOF(Fe) adsorbing $N_2$. (**e**) Alternating and (**f**) distal pathways in Al-PMOF(Fe). Reproduced with permission [87]. Copyright 2021, American Chemical Society.

MOFs can also function as photosensitizers and transfer the photogenerated electrons to active sites in other functional materials that are in contact with them. For example, Ding et al. combined N-defect thin film g-$C_3N_4$ (DF-$C_3N_4$) with nano MOF to form Nano-MOF-74@DF-$C_3N_4$ composite for enhanced photocatalytic nitrogen reduction (Figure 9a). A Z-type heterojunction was formed between Nano-MOF-74 and DF-$C_3N_4$, which clearly improved the separation efficiency of photogenerated electrons and holes (Figure 9b). The Z-type heterojunction photocatalyst structure first proposed by Bard et al. in 1979, can provide the entire system with a stronger redox ability to promote photocatalytic activity. It should be noted that another carrier transfer process of type II is competitive with the Z-type process, and the fluorescence lifetime results indicated that the Z-type is the main process. In this Z-type carrier transfer process, the photogenerated electrons in the conduction band of MOFs are firstly transferred to the valence band of DF-$C_3N_4$, then become re-excited to the conduction band of DF-$C_3N_4$, and finally anticipate the reduction of $N_2$ adsorbed on N defect sites of DF-$C_3N_4$.

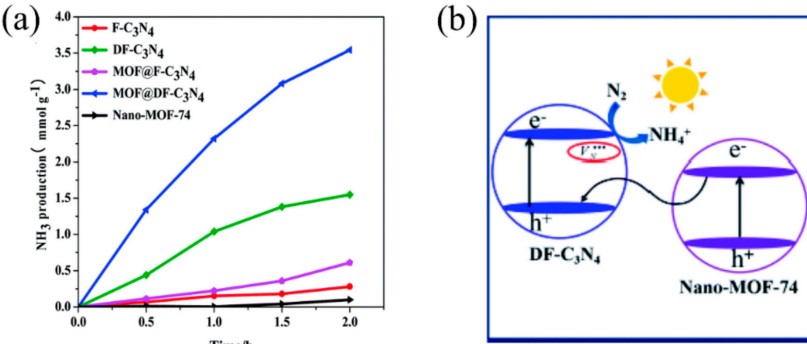

**Figure 9.** (**a**) The graph of $NH_3$ yield for F-$C_3N_4$, DF-$C_3N_4$, MOF@F-$C_3N_4$, MOF@DF-$C_3N_4$, and Nano-MOF-74 under visible light. (**b**) Z-type heterojunction of MOF@DF-$C_3N_4$. Reproduced with permission [94].

Another similar example is the MOFs composite MIL-125@$TiO_2$ with core-shell structure synthesized by Wang et al., using the post thermal solvent method (Figure 10a) [95]. The thickness of $TiO_2$ nanosheets can be controlled by the reaction time with thioacetamide. Under the synergistic effect of MIL-125 and $TiO_2$, MIL-125@$TiO_2$-2 h exhibited an activity up to 89.5 $\mu$molg$^{-1}$ h$^{-1}$ under visible light, which was much higher than the pristine MIL-125 (Figure 10b). In MIL-125@$TiO_2$, photogenerated electrons can be transferred to both the coordination unsaturated Ti sites of defect MIL-125 and $TiO_2$, and further reduce the adsorbed $N_2$ to $NH_3$.

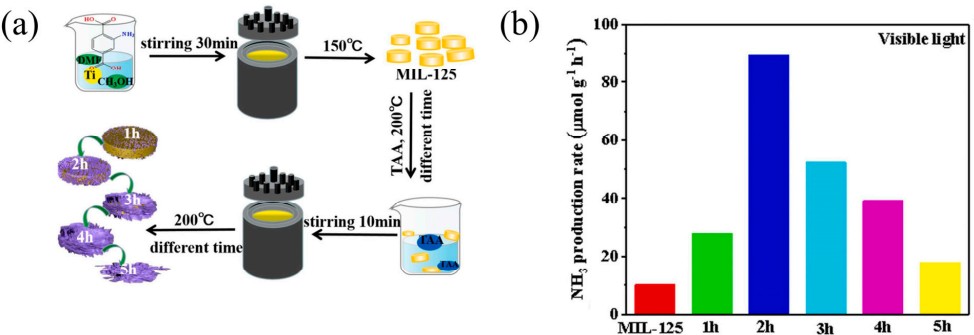

**Figure 10.** (**a**) Synthesis diagram of MIL-125 and MIL-125@$TiO_2$. (**b**) The photocatalytic nitrogen fixation efficiency of MIL-125 and MIL-125@$TiO_2$ under visible light irradiation. Reproduced with permission [95]. Copyright 2022, Elsevier.

Mxene is a new type of two-dimensional material with a graphene-like structure. It has the advantages of a large specific surface area, excellent conductivity, and abundant surface groups. Qin et al. synthesized Mxene/MOF composites (Ti$_3$C$_2$-QD/Ni-MOF, formed as type II heterojunction; QD stands for quantum dot) by combining Mxene QDs and Ni-MOF through self-assembly, which improved the photocatalytic nitrogen fixation efficiency to 88.79 $\mu$mol g$^{-1}$ h$^{-1}$ [97]. Under light irrigation, the electrons and holes in Ti$_3$C$_2$-QD were separated. Then, the photo-excited electrons were transferred to the conduction band (CB) of Ni-MOF to participate in the photocatalytic nitrogen fixation reaction. In this system, the heterojunction formed by Ti$_3$C$_2$-QD and Ni-MOF effectively promotes the separation of electrons and holes. Ni effectively adsorbed $N_2$ as an active site, and promoted the nitrogen fixation process.

MOFs can also function as porous scaffolds to support functional units, and enhance the mass transfer of $N_2$ to produce $NH_3$. For example, Chen et al. used the UiO-66 membrane as a nanoreactor to support gold nanoparticles (AuNPs) [93] and realized a direct plasma photocatalytic nitrogen reduction reaction at room temperature and ordinary pressure. Notably, in the Au@UiO-66, UiO-66 not only effectively restricts the highly

dispersed AuNPs but also ensures effective contact between AuNPs and $N_2$ molecules in the aqueous solvent. In the gas film solution reaction interface, the $N_2$ molecule can proceed straight into the Au@UiO-66 membrane. Each AuNP in UiO-66 could not only generate electrons through photo-excitation, but also promote the reduction of $N_2$ as a co-catalyst. (Figure 11a). The mass loading of gold was the critical factor that affected photocatalytic nitrogen fixation. When the loading amount of Au is 1.9 wt %, the conversion of nitrogen to ammonia reaches its highest level of 0.14 mmol $g^{-1}$ $h^{-1}$ (Figure 11b).

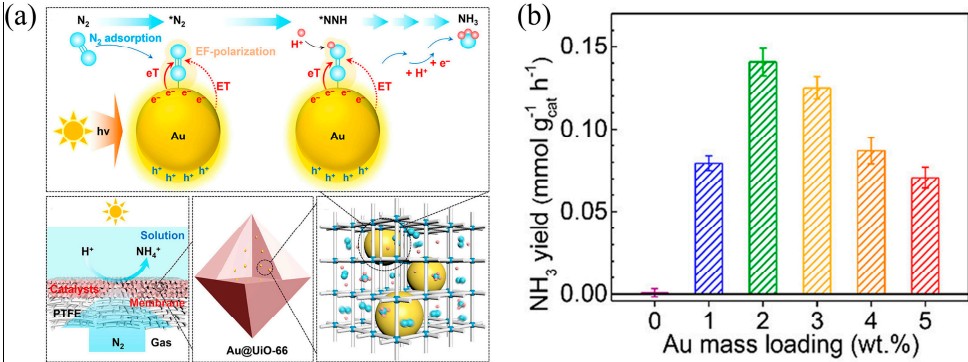

**Figure 11.** (**a**) Schematic diagram of direct plasma photocatalytic nitrogen reduction on AuNP encapsulated by UiO-66 matrix. (**b**) Direct plasma photocatalytic nitrogen reduction performances in the PiS system. Reproduced with permission [93]. Copyright 2021, American Chemical Society.

The introduced functional materials into MOFs can increase the photocatalytic nitrogen fixation activity and enhance the stability of composite materials. For example. Pan et al. reported on a graphene-embedded Ce-based UiO-66 photocatalyst (GSCe) [96]. With ultraviolet light irradiation, a breakage of benzene-C bonds inside Ce-UiO-66 formed active sites, which was evidenced by C K-edge X-ray absorption near its edge structure (reduced of C 1s → σ* at 292.0 eV) (Figure 12a). Even if the introduced graphene did not have any active sites, it could help control the activation process and hence stabilize the entire photocatalyst structure. With a graphene ratio of 0.35, GSCe reached its best performance, with a rate and apparent quantum efficiency (AQE) of 110.24 μmolL$^{-1}$ h$^{-1}$ and 9.25%, respectively (Figure 12b). The stability of GSCe even reached 7 × 24 h (Figure 12c).

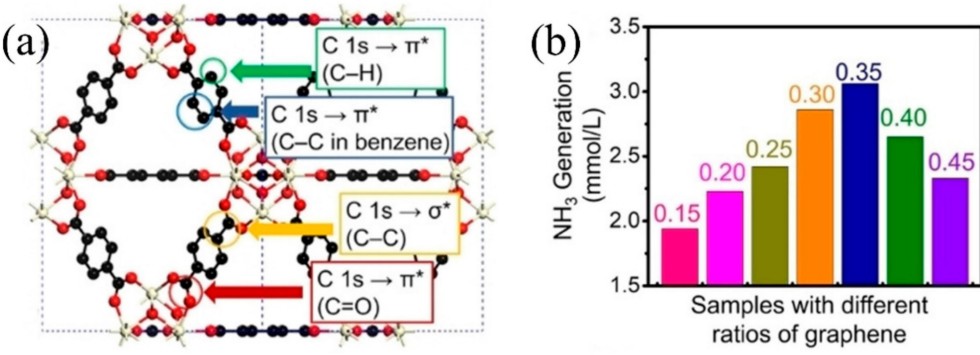

**Figure 12.** *Cont.*

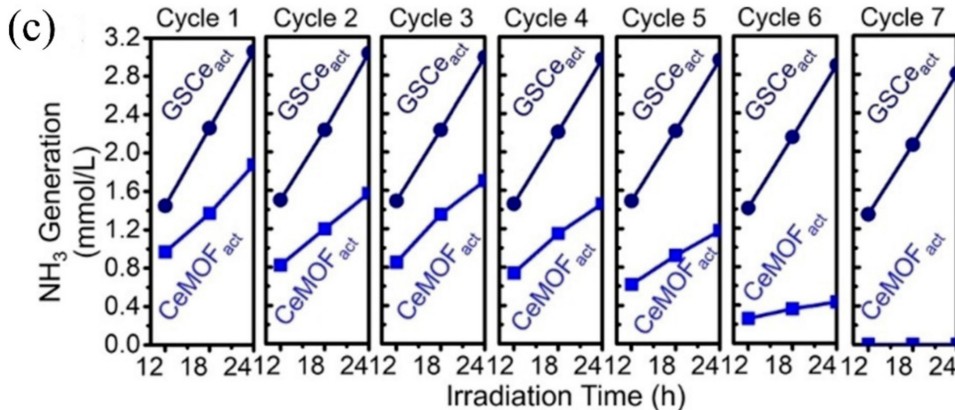

**Figure 12.** (**a**) Different bonds in Ce-UiO-66. (**b**) Photocatalytic nitrogen fixation activity with different ratios of graphene. (**c**) Recycle experiments. Reproduced with permission [96]. Copyright 2022, WILEY-VCH Verlag GmbH & Co. KgaA, Weinheim.

### 4.3. MOF Derivatives

MOFs have been proposed as a potential precursor for synthesizing nanostructured materials with porosity [73,114]. After pyrolysis in different atmospheres, MOFs can be transformed into derivatives, including metal oxides, metal carbides, metal nitrides, carbon nanostructures, etc. Those MOF derivatives can be used for electrocatalysis [115], sensing [116,117], electrode materials [118], photocatalysis [119], etc.

As an example of photocatalytic nitrogen fixation, Vu et al. synthesized an In-based hollow peanut-like photocatalyst (Ru-In$_2$O$_3$ HPNs) through pyrolysis of MIL-68-In(Ru) precursors in the air [98]. Under solar simulator (AM 1.5 G filter) irradiation, the nitrogen fixation activity of Ru-In$_2$O$_3$ HPNs reached 44.5 µmolg$^{-1}$ h$^{-1}$ (Figure 13a). This was the result of the synergistic effects of hollow structure and Ru. Under the light, electrons and holes were separated. The electrons in the conduction band were immediately captured by Ru and oxygen vacancies after migrating to In$_2$O$_3$, which activated and weakened N≡N, and finally reduced N$_2$ to NH$_3$ (Figure 13b).

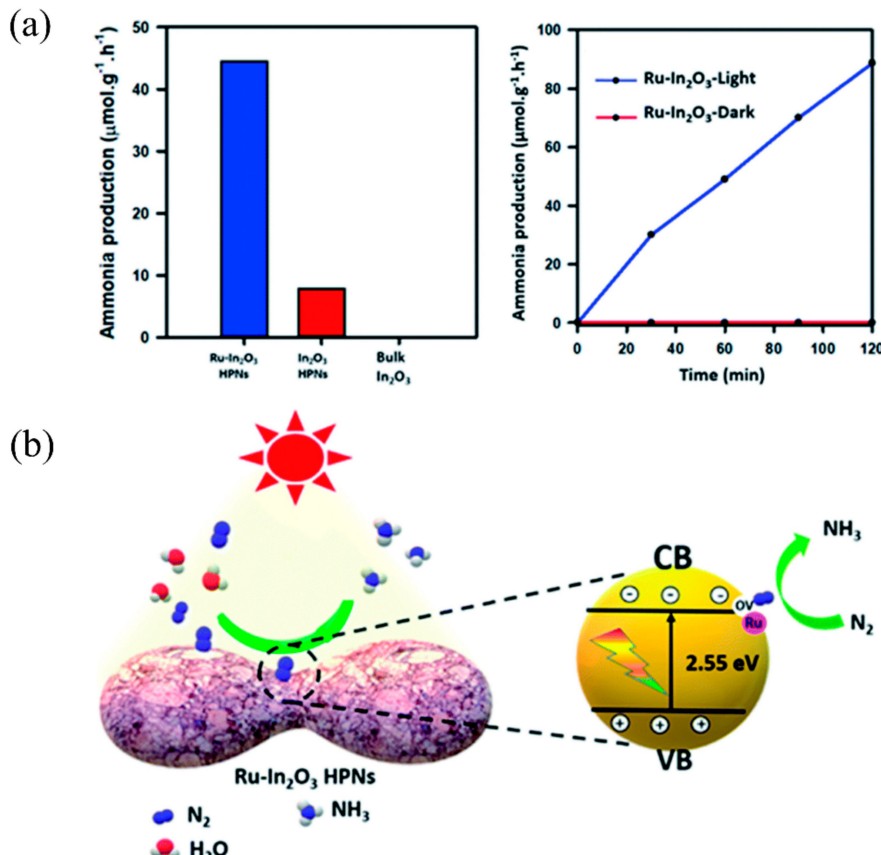

**Figure 13.** (**a**) The photocatalytic $NH_3$ formation rate of different photocatalysts under visible light. (**b**) Mechanism of photocatalytic nitrogen fixation on Ru-$In_3O_3$ HPNs. Reproduced with permission [98]. Copyright 2021, Royal Society of Chemistry.

## 5. Summary and Future Perspective

The photocatalytic fixation of nitrogen to ammonia is a significant challenge, but it is a potential technology for green ammonia production. MOF-based materials, including pristine MOFs, MOF composites, and MOF derivatives, are excellent candidates for photocatalytic nitrogen fixation. This comprehensive review highlights recent advances in applying MOF-based materials toward photocatalytic nitrogen fixation. First, this review introduced the determination method of produced ammonia. Then, the synthetic methods of MOF-based materials were summarized. Finally, the recent achievements in MOF-based materials for photocatalytic nitrogen fixation were systematically introduced. Despite considerable progress in MOF-based materials toward photocatalysis with nitrogen fixation being achieved, many challenges remain.

Improving Reliability of $NH_3$ Production. Before or during photocatalytic nitrogen fixation, the equipment, nitrogen sources, catalysts, solvents, etc., may be polluted. Therefore, in order to avoid unreliable conclusions, the following control experiments should be conducted: using inert Ar as a gas source, conducting the experiments under dark conditions, replacing protonic solvents (such as water) with aprotic solvents (such as DMF, DMSO), and conducting $^{15}N_2$ isotope labeling experiments. Furthermore, different ammonia detection methods have their advantages and limitations; therefore, suitable detection methods should be applied in different reaction systems, and several methods should be used to detect ammonia accurately.

Reporting data in photocatalytic systems accurately. Typically, catalytic rates are expressed in terms of the amount of product divided by the amount of catalyst per unit of time. Nevertheless, this parameter can be easily manipulated to the authors' convenience, especially in multicomponent systems where only a fraction of the catalyst represents

the genuine active site. For this reason, other figures of merit such as apparent quantum yield (AQY) are also useful when comparing activities between different photocatalysts. The combination of rates plus AQY seems to be a more robust way to report catalytic performances and establish fair comparisons between different materials [120].

Developing efficient photocatalysts. The coordination of unsaturated sites in MOFs can effectively improve $N_2$ adsorption/activation, which can be created with the defects generated. Therefore, creating abundant defects in MOFs while maintaining their highly porous structure are required for highly efficient photocatalytic nitrogen fixation. Moreover, broadening the light-harvesting range of MOFs can also increase the photocatalytic nitrogen fixation efficiency. Introducing light-absorbing functional groups into linkers such as amino groups, sulfhydryl groups, etc., or selecting strong photoactive ligands as linkers, such as porphyrin derivatives, phthalocyanine derivatives, naphthalene derivatives, etc., can greatly broaden the spectral responses of MOFs, thereby enhancing their light absorption efficiency. Constructing reasonable structures of MOFs to enhance the photogenerated electron and hole separation and avoid recombination can also dramatically enhance their photocatalytic nitrogen fixation efficiency. Finally, a rational combination with other functional materials (graphene, g-$C_3N_4$, Mxene, etc.) to form advanced MOF composites, or as precursors to prepare MOF-derived materials ranging from carbon-based materials to metal-based materials, is another option to enhance their photocatalytic nitrogen fixation efficiency.

Discovering photocatalytic mechanisms using in-situ characterization techniques. Traditional characterization techniques can only reflect the state of the catalyst before and after the reaction. However, the catalytic reaction requires both a steady state and a transient state. Therefore, in-situ characterization techniques have become indispensable in catalytic reactions. Compared with traditional characterization techniques, in-situ techniques are much easier to use in capturing the complex intermediates with short lifetimes to reveal the real active sites. In-situ characterization techniques, such as in-situ extended X-ray absorption fine structure (EXAFS), infrared absorption spectra, diffuse reflectance infrared Fourier transform spectroscopy (DRIFTS), thermal gravimetric analysis (TGA), XPS, Raman spectroscopy, etc., can greatly deepen our understanding of the reaction mechanisms, and further guide us to design and synthesize efficient photocatalysts. Although part of those in-situ techniques had been applied to study the photocatalytic nitrogen fixation by MOF-based materials, the research is still in its early stages.

Delving into the photocatalytic mechanism by theoretical calculation. Theoretical calculations and experiments always coexist and promote each other. MOFs with predicated structures are ideal platforms for constructing models used for theoretical calculations. With the help of theoretical calculations, important information including the electronic structures of materials, densities of states, adsorption energies of $N_2$, reaction pathways of $N_2$ and $H_2O$, changes in bond lengths and angles, etc. can be obtained to enhance our understanding of the specific relationships between the structures of photocatalysts and their corresponding performances, guiding highly efficient photocatalyst design and promoting development of the photocatalytic nitrogen fixation field.

Although MOF-based materials for photocatalytic nitrogen fixation are still in their initial stages of development, it is a rapidly growing field owing to their remarkable advantages. We hope this comprehensive review will contribute to further developments in MOF-based materials for nitrogen fixation that benefit researchers who are interested or involved in this field.

**Author Contributions:** Conceptualization, L.F., X.W. and J.G.; writing—original draft preparation, L.F. and X.W.; writing—review and editing, L.F., Q.Y., J.C., U.K., X.W. and J.G.; supervision, X.W. and J.G.; project administration, X.W. and J.G.; funding acquisition, X.W. All authors have read and agreed to the published version of the manuscript.

**Funding:** We are grateful for the financial support from the National Natural Science Foundation of China (Grant No. 22001094), Guangdong Basic and Applied Basic Research Foundation (Grant No. 2020A1515110003), and fundamental research funds of Zhejiang Sci-Tech University.

**Data Availability Statement:** Not applicable.

**Conflicts of Interest:** The authors declare no conflict of interest.

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
