# Peer review of "Achievements and Perspectives in Metal–Organic Framework-Based Materials for Photocatalytic Nitrogen Reduction"

_catalysts, doi:10.3390/catal12091005_

Round 1

Reviewer 1 Report

Achievements and Perspectives of Metal-Organic Framework- based Materials for Photocatalytic Nitrogen Reduction

Comments:

11. There are many minor grammar problems in this manuscript. I would also recommend that the authors pay particular attention to improving the grammar in their manuscript by cooperated with native English language professional. 

    2. Provide a mechanism scheme of Ammonia detection using MOFs.

3 3.  It would be good if some literatures on MOF used for ammonia detection can be added.

4 4.  Why the authors presented the ammonia detection methods in this review what is the correlation between ammonia detection with photocatalytic Nitrogen reduction using MOFs and photocatalytic nitrogen fixation?

   5. In section 3 “The mechanochemical synthesis method”, some relevant literatures on new mechanochemical synthesis of MOFs can be added.

                                i.            CrystEngComm 20,18, 2018, 2486-2490.

6.There is no discussion on ZIF type MOFs in the primary MOF section. Only in the MOF composite section, ZIF-67 composite with PMOs are reviewed.

 7.    Provide the full name of some abbreviation when used for the first time.

  8. Describe about the best techniques for the measurement of ammonia among the mixture in the catalytic process.

 9. Describe more detail about 15N2 isotope labelling experiments.

10.  How the composite of MOF and MXene improve the catalytic properties?

Reviewer 2 Report

In this review, Fan et al have reviewed recent advances and future challenges of photocatalytic nitrogen reduction using MOF-materials. Light-mediated nitrogen reduction to produce ammonia is definitely a very hot topic owing to the importance of ammonia in the manufacture of fertilizers, explosives and other nitrogen-containing chemicals. Furthermore, the use of MOF-materials in the photocatalytic nitrogen reduction reaction is still at its early stage so it is evident that this review is timely and it will be of interest for a broad audience working in the field of photocatalysis. However, it is also true that this same topic has been covered in a recent review in European Journal of Inorganic Chemistry so it is hard to see any novelty of note in the current review (https://doi.org/10.1002/ejic.202100748).

Below I suggest some changes and further discussions to increase the quality of the review:

11)      Eq 2 is not balanced. There is one missing H atom on the right side, so the equation should be rewritten using NH3 instead of NH4+

22)      Different grammatical mistakes can be found along the text:

1.       Line 53: features

2.       Line 159: have (plural)

3.       Line 226: activate (verb)

4.       Line 335: ratio

5.       Line 365: is (singular)

6.       Line 396: stabilize

7.       Line 402: adsorption

8.       Line 557: please reformulate the sentence

9.       Line 561: Delving into

10.   Line 566: angle

33)      Authors should include in their discussion the recent use of bimetallic MOFs for light-mediated N2 reduction reaction (https://doi.org/10.1039/D1CY02294F)

44)      Authors should include the use of MXene as component of MOF composites for the N2 reduction reaction (https://doi.org/10.1021/acssuschemeng.0c06388)

55)      I strongly recommend authors to provide some guidelines to accurately report data in photocatalytic systems, in this particular case for the N2 reduction reaction. Typically, catalytic rates are expressed as the amount of product divided by the amount of catalyst per unit of time. Nevertheless, this parameter can be easily manipulated to the authors’ convenience, especially in multicomponent systems where only a fraction of the catalyst represents the genuine active site. For this reason, another figures of merit, for instance apparent quantum yield (AQY), are also useful when comparing activities from different photocatalysts. The combination of rates plus AQY seems to be a more robust way to report catalytic performances and establish fair comparisons between different materials. A more detailed discussion on this can be found in a recent viewpoint from Fornasiero and collaborators (https://doi.org/10.1021/acscatal.0c01204)

Reviewer 3 Report

Page 1, line 10: Redefine coordination polymer following IUPAC terminology for MOF vs CP

Page 1, line 30: “is mainly ……………….from natural nitrogen” mainly what?

Page 2, line 50: Its Zn oxo cluster or node  “Zn4O” more than just cluster, please change

Page 2, line 50: What do you mean by “feathers” ? English need to be checked carefully throughout the MS such as Page 2, line 64:  “This review firstly summarized and compared the ammonia”

 Page 2, line 82:  please check PrOH or iPrOH

Redraw chemical equation using chemdraw applied same font and high resolution specially Eqn. 3 and 4.

Introduce table of contents for the review

Figure 2 resolution is bad, revise. Same for Figure 3 and others.

No need to add Figure 8 to the MS.

Page 15, line 444: “The reason why called Z-type is that the electron transfer process forms the shape 444 of the English letter Z in the figure” remove, it’s irrelevant

Please add paragraph to compare photocatalytic performance of MOF with those metal oxides and classical organometallic complexes

Round 2

Reviewer 2 Report

Authors have addressed all my comments. The paper is ready to accept.

Reviewer 3 Report

comments addressed